# Regulation of T Cell Activation and Metabolism by Transforming Growth Factor-Beta

**DOI:** 10.3390/biology12020297

**Published:** 2023-02-13

**Authors:** Robert J. Salmond

**Affiliations:** Leeds Institute of Medical Research at St. James’s, School of Medicine, University of Leeds, Leeds LS2 9JT, UK; r.j.salmond@leeds.ac.uk

**Keywords:** T cell activation, immunometabolism, cytokines, signaling, TGFβ

## Abstract

**Simple Summary:**

T cells are a subset of white blood cells that play essential roles in immune protection from a wide range of infectious diseases as well as cancer. By contrast, when immune responses are not controlled properly, T cells can promote damaging inflammation such as that seen in autoimmune diseases. Therefore, the activation of immune responses is tightly regulated in the body with a range of positive and negative signals involved in dictating the nature and extent of T cell responses. Transforming growth factor beta (TGFβ) is a family of proteins, termed cytokines, that have a wide range of roles in the body including an essential role in the regulation of T cell immune responses. Much research effort has gone into understanding the mechanisms by which TGFβ exerts its immune effects with a view to defining new therapies to control T cell responses in autoimmunity and cancer. This review describes recent developments in this research area with a particular focus on effects on T cell metabolism.

**Abstract:**

Transforming growth factor beta (TGFβ) receptor signalling regulates T cell development, differentiation and effector function. Expression of the immune-associated isoform of this cytokine, TGFβ1, is absolutely required for the maintenance of immunological tolerance in both mice and humans, whilst context-dependent TGFβ1 signalling regulates the differentiation of both anti- and pro-inflammatory T cell effector populations. Thus, distinct TGFβ-dependent T cell responses are implicated in the suppression or initiation of inflammatory and autoimmune diseases. In cancer settings, TGFβ signals contribute to the blockade of anti-tumour immune responses and disease progression. Given the key functions of TGFβ in the regulation of immune responses and the potential for therapeutic targeting of TGFβ-dependent pathways, the mechanisms underpinning these pleiotropic effects have been the subject of much investigation. This review focuses on accumulating evidence suggesting that modulation of T cell metabolism represents a major mechanism by which TGFβ influences T cell immunity.

## 1. Introduction

Appropriate regulation of T cell responses balances the activation of protective immunity with the maintenance of self-tolerance. Circulating naïve T cells can be retained in a quiescent, non-functional state for decades in humans. However, within 24 h of encountering their cognate antigens presented by activated antigen-presenting cells, T cells are triggered to undergo rapid cell growth, before undergoing clonal expansion and effector differentiation. Furthermore, naïve T cells have the capacity to give rise to a number of effector T cell lineages, characterized by possession of distinct effector mechanisms, and long-lived memory cells. This functional diversity and capacity for plasticity enable T cells to participate in immune responses to classes of pathogens that require different mechanisms for effective clearance (e.g., viruses vs. multicellular parasites) in a wide range of anatomical locations. The ability to prevent inappropriate activation of T cells and switch off ongoing T cell responses is equally important for the maintenance of immune health. By contrast, immune tolerance mechanisms can be subverted in cancer preventing effective T cell responses to tumors.

Integration of antigen, co-stimulatory and cytokine receptor signals with environmental cues is key to the regulation of T cell responses. Furthermore, regulation of fundamental aspects of cellular metabolism is central to T cell fate and the outcome of immune responses. Transforming growth factor beta (TGFβ) signals impact upon every stage of the T cell life cycle, from thymic development through to activation and differentiation of effector cells, and generation of long-lived T cell memory. Much research effort has been expended in determining the molecular mechanisms underpinning the pleiotropic effects of TGFβ in the immune response and in the development of approaches to target TGFβ in therapeutic settings. Several excellent recent reviews cover much of this key material [1,2,3]. In the current work, we focus on metabolic mechanisms of TGFβ action on T cell responses.

## 2. Overview of T Cell Metabolism

Naïve T cells are quiescent and uptake low levels of glucose and amino acids from the environment in order to fuel a catabolic metabolism that maintains homeostatic levels of ATP production and biosynthesis. By contrast, upon activation, T cells undergo metabolic reprogramming in order to fuel a dramatic increase in energetic demands. Whilst naïve T cells predominantly utilize mitochondrial oxidative phosphorylation (OXPHOS) for ATP synthesis, activated T cells substantially upregulate expression of nutrient receptors and use glycolysis, glutaminolysis and lipid synthesis pathways to support the demands of growth, activation and proliferation. Effector T cell responses are disrupted upon deletion of genes encoding key nutrient transporters such as glucose transporter 1 (GLUT1) [4] or amino acid transporters Slc7a5 [5] and Slc1a5 [6], demonstrating the reliance of T cell activation upon uptake of extracellular nutrients. A dependence on glycolytic metabolism is a feature of both non-malignant and transformed proliferative cells and enables T cells to fuel the pentose phosphate pathway (PPP). Nicotinamide adenine dinucleotide phosphate (NADPH) produced via the PPP is rate-limiting for many biosynthetic processes and the production of cellular biomass [7]. In both effector and memory T cells, glucose metabolism, via glycolysis and the TCA cycle, is also linked to *de novo* fatty acid synthesis [8,9]. Furthermore, in activated T cells, mitochondrial metabolism is essential for both macromolecule biosynthesis and production of reactive oxygen species that drive nuclear factor of activated T cells (NFAT) activation and interleukin-2 (IL-2) production [10].

T cell antigen receptor (TCR), CD28 costimulatory and cytokine signals contribute to activation of metabolic signalling. In particular, activation of mechanistic target of rapamycin complex 1 (mTORC1) and Myc signalling is important for T cell metabolic reprogramming. mTOR and Myc pathways are required for upregulation of nutrient transporters and subsequent glycolytic and amino acid metabolism [11,12,13,14,15]. Additional transcriptional regulators including hypoxia-inducible factor-1α (HIF1α) [16,17], estrogen-related receptor-α [18] and sterol regulatory element-binding proteins (SREBPs) [19] also contribute to metabolic remodelling in T cells. For a more in-depth overview of the regulation and role of metabolic signalling and reprogramming in T cell responses, recent reviews are recommended [20,21,22,23]. 

## 3. Fundamentals of TGFβ Signalling

TGFβ is a family of 3 cytokines that have diverse physiological functions. TGFβ1 (hereafter simply referred to as TGFβ) is the principal immune-associated isoform. TGFβ can be produced by many cells within the body and is assembled as a latent complex comprising two copies of an N-terminal latency-associated peptide (LAP) and the C-terminal active cytokine. This inactive complex can be linked to latent TGFβ-binding proteins (LTBPs) or membrane-bound proteins such as glycoprotein A repetitions predominant (GARP). LTBPs and GARP target latent TGFβ to the extracellular matrix and plasma membrane, respectively, whilst activation of TGFβ, via removal of LAP, is dependent upon interaction with integrins. Integrins αVβ6 and αVβ8 are essential for the activation and immune functions of TGFβ (reviewed in [24]).

Active TGFβ binds to cell membrane heterotetrameric TGFβ type I and type II serine/threonine kinase receptors (TGFBR1 and TGFBR2). Upon ligand binding, TGFBR2 phosphorylates TGFBR1 that subsequently phosphorylates small mothers against decapentaplegic (SMAD) 2 and SMAD3. Following phosphorylation, these receptor-associated SMAD proteins form a complex with SMAD4 that translocates to the nucleus to regulate target gene expression. Inhibitory SMAD7 and SKI-like proto-oncogene (SKIL) act as feedback inhibitors of SMAD signalling. Furthermore, TGFβ-induced SMAD-independent pathways include TGFβ-activated kinase (TAK) 1-dependent activation of Jun N-terminal kinase (JNK) and p38 mitogen-activated protein kinase (MAPK) pathways, and phosphoinositide 3-kinase (PI3K) signaling (reviewed in [25]). Gene-targeting approaches have determined an essential role for TGFβ and its downstream signalling components in the immune response. Thus, mice lacking TGFβ1 or mice with T cell-specific deletion/inhibition of TGFBR2 or combined SMAD2/3-deficiency develop an early onset, multifocal and fatal autoimmune disease [26,27,28,29]. Furthermore, genetic deficiency of TGFβ1 results in severe inflammatory and autoimmune phenotypes in humans [30].

## 4. TGFβ Modulation of T Cell Activation and Metabolism

### 4.1. T Cell Differentiation Is Accompanied by and Dependent upon Metabolic Reprogramming

Following activation by TCR and costimulatory signals, and in the presence of polarizing cytokines, naïve CD4^+^ T cells differentiate to a number of effector T helper (Th) lineages that are defined by expression of specific transcription factors and effector cytokines, summarized in Figure 1. Briefly, IL-2 and IL-12 drive polarization to a Th1 phenotype characterized by expression of lineage-defining transcription factor T box expressed in T cells (Tbet) and effector cytokines such as IL-2, tumor necrosis factor (TNF) and interferon-γ (IFNγ). By contrast, IL-4, IL-25 and IL-33 drive the differentiation of Th2 cells that express GATA binding protein 3 (GATA3) and effector cytokines IL-4, IL-5 and IL-13. Importantly, TGFβ suppresses expression of both Tbet [31] and GATA3 [32], thereby inhibiting Th1 and Th2 differentiation. Instead, TGFβ synergizes with IL-6 to favor differentiation of Th17 cells that express retinoic acid-related orphan receptor γ (RORγ) and produce high levels of IL-17 [33] or with IL-4 to induce IL-9-secreting Th9 cells [34,35,36]. TGFβ-dependent Th17 cells contribute to immune defense at barrier sites such as the gut but are also involved in the pathogenesis of autoimmune and inflammatory diseases. Th9 cells are thought to be involved in responses to parasitic infections and in allergies. The role of TGFβ in the differentiation of IL-22-secreting T cells has been controversial. Data from human studies suggest that Th22 cells are a stable and distinct helper subset [37], whilst in vitro studies suggest that TGFβ limits the polarization of naïve CD4^+^ T cells to a Th22 phenotype [38]. By contrast, more recently, evidence has shown that TGFβ promotes IL-22 production by Th17 cells in vitro and in vivo [39]. Furthermore, a positive role for TGFβ signalling in the licensing and differentiation of follicular helper T cells (Tfh), which localize to B cell follicles of secondary lymphoid tissue, has been reported [40,41]. During CD4^+^ T cell differentiation, TGFβ suppresses expression of the chromatin organizer and transcription factor special AT-rich sequence-binding protein-1 (SATB1) via SMAD3-dependent signals, which in turn enables enhanced expression of inducible co-stimulator (ICOS) that promotes Tfh differentiation [41,42]. Finally, in combination with IL-2, TGFβ promotes the differentiation of naïve CD4^+^ T cells to an induced regulatory T cell (iTreg) phenotype characterized by suppressive function and expression of forkhead box P3 (FOXP3) [43,44]. Similarly, development of “natural” (n)Tregs depends upon TGFβ signaling in the thymus [45]. Treg cells are essential for the maintenance of immune tolerance but may also contribute to immune dysfunction in cancer.

CD8^+^ T cells may also adopt phenotypes analogous to their CD4^+^ T cell counterparts. The best-studied effector CD8^+^ T lymphocytes, termed cytotoxic T lymphocytes (CTLs) or Tc1, mediate target cell killing via the production of cytolytic granules and perforin, and secrete effector cytokines such as IFNγ and TNF. Under cytokine polarization conditions that give rise to Th2, Th9 and Th17 populations, CD8^+^ Tc2, Tc9 and Tc17 effector populations have been described [46]. Of note, TGFβ-dependent CD4^+^ and CD8^+^ effector T cell lineages display phenotypic plasticity frequently, e.g., Th17 cells can convert to Th1 and Treg phenotypes in vivo [47].

The distinct T cell subsets differ not only in their effector mechanisms but also in their metabolic phenotypes. It is clear that (i) metabolism influences effector polarization and (ii) the cytokines that drive polarization to distinct effector lineages influence the metabolic phenotype of those cell populations. Thus, iTregs typically have lower expression of GLUT1 and glycolytic metabolism as compared to effector Th1, Th2, Th9 and Th17 lineages [48,49]. Consistent with the idea that metabolism determines effector phenotypes, enforced expression of GLUT1 in transgenic mouse T cells elevates glucose uptake and favors differentiation of Th1 and Th2 cells, at the expense of Tregs [48], whereas effector T cells but not Treg cell numbers are impaired in the absence of GLUT1 [4]. By contrast, iTregs have increased mitochondrial membrane potential (MMP) and higher rates of lipid oxidation compared with effector lineages [16,48], whilst inhibition of carnitine palmitoyl-transferase-1 and lipid oxidation selectively inhibits iTreg differentiation [48]. Th17 differentiation conditions induce the upregulation of acetyl-CoA carboxylase 1 (ACC1), which mediates glucose-dependent de novo fatty acid synthesis, whilst deletion of ACC1 diverts Th17 cells to a Treg phenotype [8]. Furthermore, recent studies using an in vivo CRISPR screen approach indicate a key role in the regulation of metabolic pathways in the Tfh versus Th1 fate decision. In this study, HIF-1α was shown to suppress Myc and mTORC1-dependent pathways, whilst degradation of HIF1α was selectively required for Tfh differentiation [50]. 

### 4.2. TGFβ Modulates Treg Metabolism via FoxP3-Dependent and Independent Effects

As described above, it is clear that TGFβ has context-dependent effects on T cell differentiation and metabolism. Thus, Th9, Th17 and iTreg differentiation all involve TGFβ signaling but result in distinct effector and metabolic profiles. The balance of Th17 vs. Treg differentiation is regulated through inhibitory pathways that restrain alternative differentiation programs. Thus, IL-2/signal transducer and activator of transcription (STAT) 5 signals suppress Th17 but promote Treg differentiation, whilst IL-6/STAT3 signals impede Tregs but support Th17 differentiation [51,52]. Evidence also suggests differential requirements for downstream SMAD family members and other TGFBR-dependent signals in Treg vs Th17 differentiation [53]. 

It is likely that TGFβ-dependent upregulation of FoxP3 selectively in Tregs is an important mechanism underpinning the distinct metabolic phenotype of these immunosuppressive populations. Deletion of FoxP3 in mouse thymic Tregs resulted in upregulation of genes involved in glucose metabolism including those encoding GLUT1 and hexokinase 2 (Hk2) [54]. By contrast, ectopic expression of FoxP3 in CD4^+^ T cells induced expression of genes involved in lipid oxidation and downregulated glycolytic genes. Therefore, expression of FoxP3, induced either by transgenic overexpression or naturally by TGFβ, directly results in decreased glycolytic flux and increased lipid oxidation and oxygen consumption rates (OCR), a proxy for OXPHOS [54,55]. Using a FoxP3-reporter mouse strain, Howie and colleagues reported that CD4^+^ T cells exposed to TGFβ that do not upregulate FoxP3 have reduced maximal respiration compared with FoxP3^+^ cells cultured in the same conditions, yet have the higher spare respiratory capacity (SRC) as compared to T cells activated in the absence of TGFβ [55]. Chromatin immunoprecipitation experiments determined that FoxP3 directly binds and suppresses *Myc* expression in mouse Tregs, underpinning the inhibitory effects on glycolytic metabolism [56]. As will be discussed below, Myc is also targeted directly by TGFβ-induced SMAD-dependent pathways and may represent a major mechanism by which TGFβ influences T cell metabolism.

Priyadharshini and colleagues reported interesting differences in the metabolic profiles of nTregs and iTregs. Following activation with CD3 and CD28 antibodies for 3 d in vitro, despite expressing high levels of FoxP3 and in contrast to iTregs, nTregs displayed comparable levels of glycolysis and glutaminolysis to effector CD4^+^ T cells [57]. Nonetheless, when nTregs were activated in the presence of TGFβ, glycolytic function was suppressed. Mechanistically, TGFβ served to inhibit glycolysis in nTregs via inhibition of mTOR activity and suppression of GLUT1, GLUT3 and Hk2 mRNA expression. Thus, Treg populations have a degree of metabolic plasticity that is affected by TGFβ during development and differentiation and also by exposure of mature Tregs to TGFβ. It is important to note that Tregs are considerably more proliferative in vivo than in vitro, and their metabolic phenotypes likely reflect these differences. Thus, amino acid uptake and TCR signals sustain higher levels of mTOR activity in Tregs as compared to naïve T cells under homeostatic conditions in vivo [58,59]. Furthermore, Tregs are able to employ a combination of glycolytic metabolism and fatty acid synthesis that enables their selective expansion within challenging tumor microenvironments [60].

### 4.3. Modulation of Glycolytic Metabolism during Th9 Differentiation by TGFβ

Wang et al. reported that Th9 populations, induced by TGFβ and IL-4, had higher glycolytic capacity than Th1, Th2, Th17 and iTreg cells [49]. This metabolic phenotype was associated with lower levels of expression of the histone deacetylase sirtuin (SIRT) 1 in Th9 cells as compared to Th1, Th2 or Th17 cells and much lower than Tregs. TGFβ signaling was the main driver of SIRT1 downregulation in Th9 cells, and this effect was enhanced by IL-4. In contrast to the effects of TGFβ in driving IL-9 production that is strictly SMAD2 and SMAD4-dependent [49,61], inhibition of TAK1 reversed the effects of TGFβ on SIRT1 expression [49]. SIRT1-deficient T cells had increased glycolytic activity and Th9 responses, indicating that SIRT1 opposes Th9 differentiation and glycolytic metabolism and that TGFβ acts to limit this effect. In the absence of SIRT1, elevated mTOR and HIF1α signals drove Th9 polarisation, whilst inhibition of glycolysis using the glucose analogue 2-deoxyglucose blocked these phenotypes [49]. Together, the metabolic phenotypes of Treg and Th9 cells exemplify how TGFβ signals have context-dependent and sometimes opposing effects on T cell metabolism; in the case of Tregs, TGFβ opposes glycolytic metabolism via direct inhibitory signalling and upregulation of FoxP3, yet is required to support glycolytic metabolism via downregulation of SIRT1 in the context of Th9 polarizing conditions.

### 4.4. Effects of TGFβ on Effector CD4^+^ T Cell Metabolism

The effects of TGFβ on metabolic reprogramming during initial activation and differentiation of CD4^+^ T cells have been well documented. Less is known of the effects on previously activated effector T cells. Dimeloe and colleagues reported that exposure of human effector/memory CD4^+^ T cells to TGFβ for 16 h in vitro decreased basal OCR, with a more modest effect on extracellular acidification rates (ECAR), a measure of lactate secretion [62]. A more prolonged culture (72 h) demonstrated that TGFβ increased MMP and SRC. These effects of TGFβ were linked to a direct association of SMAD proteins with mitochondria and inhibition of ATP synthase activity. Importantly, inhibition of mitochondrial activity by both recombinant and tumor-derived TGFβ resulted in inhibition of CD4^+^ T cell effector cytokine production [62]. Consistent with the effects of TGFβ on mitochondria, a previous report determined that exposure to TGFβ transiently increased MMP in peripheral blood leukocytes from systemic lupus erythematosus patients [63].

### 4.5. Impact of TGFβ on CD8^+^ T Cell Activation and Metabolism

Seminal studies by the Massagué group determined that TGFβ switches off many of the key genes involved in CTL effector function [64]. Thus, TGFβ limits TCR-induced expression of *Il2, Ifng*, *Gzmb*, *Gzma* and *Prf1* by SMAD-dependent pathways, impeding effective CD8^+^ T cell responses in tumors. TGFβ inhibition of CTL activation is most effective under conditions of weak antigenic stimulation [65,66], reflecting the key role of TGFβ in the maintenance of T cell tolerance to low affinity self-antigen. Recent data indicate that inhibition of TCR-induced metabolic reprogramming is a further mechanism by which TGFβ limits CD8^+^ T cell responses (Figure 2). *MYC*/*Myc* has been shown to be a target of TGFβ signalling in a variety of cell types in humans and mice, including T cells, and negative regulation of Myc expression has been linked to the anti-proliferative effects of TGFβ [67,68,69]. Consistent with these earlier studies, RNA-sequencing (RNA-Seq) analysis of OT-I TCR transgenic CD8^+^ T cells demonstrated that TGFβ limited TCR-induced upregulation of Myc expression and subsequent Myc-dependent transcriptional programmes [70]. TGFβ-treated TCR-stimulated CD8^+^ T cells had reduced mRNA levels of genes encoding glycolytic enzymes as well as glucose and amino acid transporters, whilst levels of glycolysis and protein synthesis were greatly impaired as compared to control cells [70], reflecting the essential role for Myc in metabolic reprogramming [11,15]. In this report, there was no apparent inhibitory effect of TGFβ on CD8^+^ mTORC1 activity, in contrast to reports in other cell systems [57,71,72].

### 4.6. Regulation of T Cell Exhaustion

T cell exhaustion under conditions of chronic antigenic stimulation is characterized by the acquisition of a dysfunctional phenotype, associated with reduced capacity to produce effector cytokines and elevated expression of inhibitory receptors such as programmed death-1 (PD-1), lymphocyte activation gene 3 (LAG3) and T cell immunoglobulin and mucin-containing protein 3 (TIM3) [73]. Adoption of the exhausted phenotype is progressive and, at least for early progenitor exhausted T cells (Tpex), is reversible through blockade of immune checkpoints such as PD-1. Evidence indicates that TGFβ signals are important in the regulation of metabolism in T cell exhaustion. Gabriel and colleagues reported that, during the early stages of chronic lymphocyte choriomeningitis virus (LCMV) infection in mice, TGFβ suppressed mTOR activity in CD8^+^ Tpex and reduced accumulation of terminally exhausted cells [72]. TGFβ-dependent inhibition of mTOR during LCMV infection was associated with enhanced mitochondrial mass and maintenance of metabolic fitness in Tpex cells but reduced effector cell function. Similar results were reported by Hu et al. who reported that, in chronic LCMV models, TGFBR2 knockout P14 TCR transgenic T cells preferentially gave rise to effector-like cells, whilst TGFβ signals maintained a PD-1^+^ T cell factor 1^+^ stem-like T cell population [74]. Furthermore, these authors demonstrated that loss of TGFB2 signalling resulted in enhanced transcription of mTOR-associated and glycolytic genes in antigen-specific T cells. Recent data support a role for TGFβ in human T cell exhaustion. Using an in vitro model of T cell exhaustion, Saadey and colleagues demonstrated that TGFβ limited T cell terminal differentiation and apoptosis under the condition of strong, chronic TCR stimulation [75]. 

### 4.7. Effects of TGFβ on Memory T Cell Formation and Metabolism

Studies from the Ahmed group determined, that in contrast to the effects on effector T cell activation, suppression of mTOR activity results in enhanced CD8^+^ T cell memory responses [76]. Consistent with this, resting memory T cells revert to a catabolic metabolic phenotype, relying on OXPHOS and fatty acid oxidation. Memory cells typically have enhanced mitochondrial mass, as compared to naïve T cells, and upon re-activation upregulate glycolysis and OXPHOS to fuel rapid proliferation and cytokine production [77,78]. Nonetheless, given the number of distinct cell populations and phenotypes, it is apparent that memory T cell metabolism is as varied and flexible as their effector T cell counterparts. General features of memory T cell metabolism have been reviewed in more depth in recent articles [79,80]. 

TGFβ signaling has been linked to the development and retention of CD4^+^ and CD8^+^ memory T cells [81,82,83]. Briefly, during the initial CD8^+^ T cell response to infection, serum TGFβ levels are increased and are important in controlling the magnitude of the effector response by reducing Bcl-2 expression and inducing selective apoptosis of short-lived effector cells [81,82]. By contrast, TGFBR2 expression is required for efficient T cell anti-viral recall responses, indicating that TGFβ is required to maintain memory T cell function [83]. In the past two decades, evidence has accumulated showing that large numbers of non-circulating tissue-resident memory T cells (Trm) localize to epithelial and mucosal tissues [84]. Importantly, in the absence of TGFBR2 expression, differentiation of skin-infiltrating CD8^+^ T cells to Trm is impeded [85]. Trm cells express a range of integrins, chemokine receptors and cell surface antigens that are important for their tissue retention, including CD69, CD103, CD49b and CXCR4 [84]. TCR signals combined with TGFβ that drives SMAD-dependent upregulation of CD103 (integrin αEβ7) are required for Trm development [86,87]. Furthermore, Tbet-expressing Type 1 Tregs are important for Trm accumulation through their capacity to regulate the bioavailability of active TGFβ in tissues through their expression of integrin αVβ8 [88]. The metabolic environment of tissues, and in particular oxygen tension, likely contributes towards Trm differentiation. Thus, hypoxia and TGFβ synergistically induce human T cell differentiation to a Trm-like phenotype in vitro [89]. Interestingly, in these experiments, TGFβ induced upregulation of metabolic transcription factor *HIF1A* expression, whilst pharmacological stabilization of HIF1A could partially mimic the effects of hypoxia on Trm differentiation [89]. Whilst the functions of TGFβ signals in the establishment and maintenance of Trm have been defined, the role of TGFβ in Trm metabolism is less understood. Trm cells have a distinct metabolic profile and are dependent upon lipid uptake and transport mediated via the lipid chaperones fatty acid binding protein 4 (FABP4) and FABP5 [90]. Analysis of a published RNA-Seq dataset [70] demonstrated that in contrast to the strong upregulation of *Itgae* (encoding CD103), *Fabp5* expression was downregulated by TGFβ during the early stages (24 h) of CD8^+^ T cell activation. These data imply that TGFβ might not be directly involved in establishing Trm metabolic phenotypes; however, further studies will be required to test these hypotheses directly. 

### 4.8. TGFβ Modulation of Metabolism during T Cell Anti-Tumour Immunity

High levels of TGFβ within the tumor microenvironment are frequently associated with loss of effector T cell function and immunosuppression [2,3]. As a consequence, targeting the inhibitory effects of TGFβ on T cells has long been regarded as a potential immunotherapeutic approach for cancer. Furthermore, evidence suggests that TGFβ has profound effects on cancer cell metabolism and, in some situations, may promote metabolic changes that precipitate cancer progression (reviewed in [91]). Here, we briefly describe approaches to reverse the metabolic effects of TGFβ on T cells in the context of anti-tumor responses. 

The ectonuclease CD73, encoded by *NT5E* in humans, is a major target for TGFβ in T cells. Along with CD39, CD73 converts ATP/AMP to adenosine that acts to suppress immune responses via adenosinergic pathways. Adenosine impacts upon tumor-infiltrating lymphocytes’ (TIL) metabolism through suppression of mTOR signalling [92]. Chatterjee and colleagues reported that Th17 cells polarized in the presence of TGFβ had low levels of IFNγ production, a propensity to convert to a Treg phenotype, and poor anti-tumor activity in vivo [93]. This was associated with, and dependent upon, TGFβ-induced CD73 expression as CD73-deficient T cells polarized under the same conditions did not have the same phenotype. Similarly, wild-type Th17 cells polarized in the presence of IL-1β, rather than TGFβ, had reduced CD73 expression and improved in vivo anti-tumor activity [93]. Of note, apoptotic Tregs induced by oxidative stress within tumors were shown to have immunosuppressive activity dependent upon ATP release and adenosine production via CD39 and CD73 [94]. Furthermore, high levels of TGFβ within the tumor microenvironment sustain CD73 expression in tumor-infiltrating T cells, resulting in high levels of adenosine-dependent immunosuppression and subsequent resistance to anti-CD137 therapies [95]. These data suggest that the TGFβ–CD73–adenosine axis may serve as an attractive target to improve cancer immunotherapies. In this regard, deletion of adenosine receptor expression in chimeric antigen-receptor (CAR) T cells and the use of receptor antagonists have been suggested as approaches to improve the outcome of T cell-based cancer immunotherapies [96,97,98,99,100,101]. Furthermore, a recent Phase 1 study reported that a CD73-TGFβ-targeting bifunctional antibody, dalutrafusp alfa, was well-tolerated in patients with advanced solid tumors [102]; further, larger studies will be required to demonstrate clinical efficacy. 

As described above, TGFβ acts to suppress TCR-induced Myc expression and downstream metabolic reprogramming. Stephen and colleagues reported that expression of transcriptional regulator FoxP1 was required for TGFβ-dependent inhibition of TCR-induced Myc and c-Jun expression and subsequent T cell activation [103]. FoxP1 was shown to be a necessary component of a transcriptionally repressive TGFβ-induced SMAD2/3 complex that bound to promoter regions and limited transcription of target genes. Levels of FoxP1 expression were shown to be elevated in TILs as compared to non-tumor-associated T cells in ovarian cancer patients. Furthermore, adoptive transfer of tumor-primed FoxP1-deficient T cells enhanced survival in mouse models of ovarian cancer and sarcoma, relative to wild-type T cell transfer [103]. More recently, high levels of expression of FoxP1 in primary human breast cancer TILs were shown to be associated with poor prognosis [104]. Together, these data suggest that FoxP1 may represent a useful target to overcome TGFβ-mediated T cell inhibition in cancer.

## 5. Conclusions and Future Prospects

As described in this review, TGFβ is a key regulator of T cell activation, differentiation, survival, exhaustion and memory. We suggest that the regulation of T cell metabolism is central to these effects, yet many outstanding questions remain. To address these, it will be vital to assess the impacts of TGFβ on T cell metabolism in vivo. It is worthy of note that many studies describing immunometabolic phenotypes have concentrated on in vitro-generated T cell subsets, which do not necessarily reflect in vivo phenotypes. For example, standard in vitro media conditions tend to favour the adoption of a glycolytic metabolism by T cells that do not faithfully reflect the in vivo situation [105]. Indeed, recent reports show that TGFβ-dependent Th17 cells generated under OXPHOS-promoting media conditions [106] better reflect in vivo phenotypes [107]. It is hoped that new approaches that allow single-cell analyses of metabolic phenotypes [108,109] will help decipher the roles of TGFβ in T cell metabolism in vivo. A further key question is how to disentangle the effects of TGFβ on metabolism from effects on other aspects of T cell biology. Many studies have relied on deletion of the TGFβ receptor in T cells and therefore impact upon all downstream pathways and targets. More sophisticated genetic approaches may be required to selectively assess the contribution of metabolic effects of TGFβ on T cell responses in vivo. This might include deletion of downstream TGFβ target genes or modification of target gene loci to selectively abrogate TGFβ/SMAD-dependent regulation. 

In recent years, the efficacy of targeting TGFβ in combination with other cancer immunotherapy modalities has been assessed. A number of clinical trials have been established to test the impact of combined TGFβ inhibition and immune checkpoint inhibitors in a range of treatment-refractory cancer types (reviewed in [110]). Importantly, a recent phase 1 trial demonstrated that dual PD-L1 and TGFβ blockade safely enhanced anti-tumor immunity in a cohort of patients with newly diagnosed human papilloma virus (HPV)-unrelated head and neck squamous cell carcinoma [111]. Furthermore, a recent preclinical study demonstrated that agonists of the TGFβ family member bone morphogenetic protein 4 could oppose TGFβ-induced effector dysfunction in human T cells and improve response to PD-1 immune checkpoint blockade in mice, suggesting alternative approaches to blocking the inhibitory effects of TGFβ on anti-cancer immunity [75]. Recent evidence suggests that combining TGFβ blockade with adoptive T cell therapies may also improve cancer outcomes. Thus, studies have reported that combining receptor tyrosine kinase like orphan receptor 1 (ROR1)-specific CAR T cells with the TGFBR kinase inhibitor SD-208 enhanced clearance of tumors in a mouse model of triple-negative breast cancer [112], whilst CRISPR-mediated deletion of TGFBR2 improved the efficacy of anti-mesothelin CAR-T cells in patient-derived pancreatic cancer xenograft models [113]. Furthermore, in-human phase I trials determined the feasibility of targeting TGFBR in prostate cancer-directed CAR T cells, using a dominant-negative receptor approach [114]. Alongside these established approaches to block TGFβ signaling, it is hoped that an improved understanding of TGFβ-dependent targets, including metabolic pathways, may lead to improved or more selective targets for T cell-based immunotherapies.

## Figures and Tables

**Figure 1 biology-12-00297-f001:**
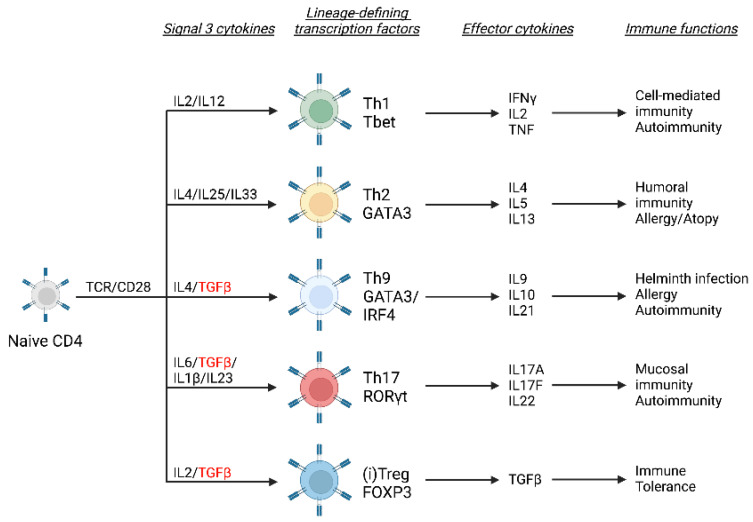
Role of TGFβ in differentiation of CD4^+^ T helper cells. Upon activation through TCR, costimulatory CD28 and cytokine-driven signaling, naïve CD4^+^ T cells adopt specialized Th cell phenotypes. Expression of lineage-defining transcription factors and effector cytokines enables distinct Th subsets to function in diverse immune responses, as depicted. Image created in Biorender.com.

**Figure 2 biology-12-00297-f002:**
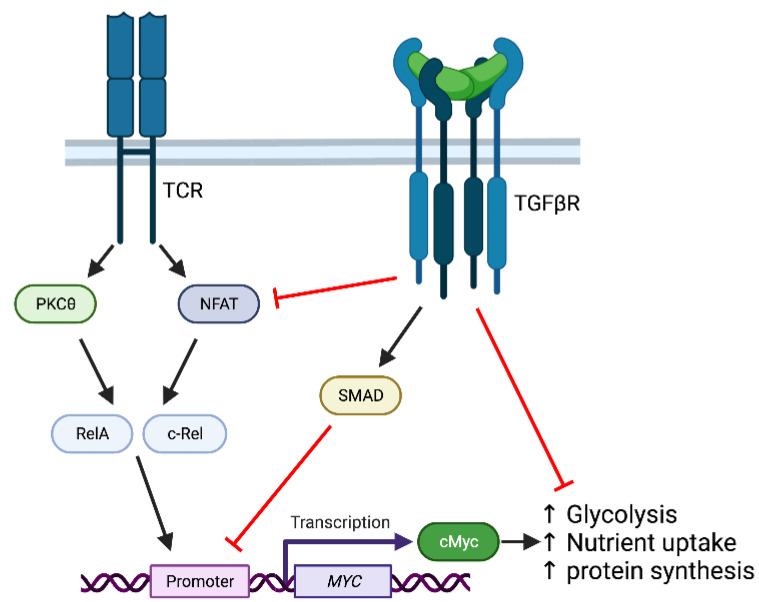
TGFβ signalling inhibits TCR-dependent Myc upregulation and metabolic reprogramming in CD8^+^ T cells. TCR signaling triggers activation of PKCtheta and NFAT that regulate RelA nuclear import and c-Rel gene expression. RelA and c-Rel regulate transcription of *Myc* that in turn facilitates upregulation of glycolytic metabolism, nutrient uptake and protein synthesis [11,15]. TGFβ receptor (TGFβR) signals inhibit NFAT activation [66], whilst SMADs bind directly to the promoter region and suppress transcription of *Myc* [67,68]. Image created in Biorender.com.

## Data Availability

Not applicable.

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
