# Peer review of "Regulation of T Cell Activation and Metabolism by Transforming Growth Factor-Beta"

_biology, 2023, doi:10.3390/biology12020297_

Round 1
Reviewer 1 Report
Dear Author
Your review report entitled " regulation of T cell activation and metabolism by TGF beta" is a well designed and well-written manuscript which highlights the potentials of TGF beta to be considered as an immunotheraputic target in the field of inflammatory diseases as well as cancer. I would suggest few minor points as follows to improve the sounds of some parts;
1. Section 4.7: lines 353-356: Please provide more citations or explanation for the potential of TGF-CD73-Adenosine axis in relation to CAR-T cells in cancer immunotherapy.
2. Section 4.7: line 367: Please explain in more details and in point of mechanistic view the relation between FoxP3 and TGF beta in cancer immunity.
Sincerely
Author Response
I'd like to thank the reviewer for their useful comments. I have adjusted the manuscript as follows (comments in bold, my replies in italics:
Your review report entitled " regulation of T cell activation and metabolism by TGF beta" is a well designed and well-written manuscript which highlights the potentials of TGF beta to be considered as an immunotheraputic target in the field of inflammatory diseases as well as cancer. I would suggest few minor points as follows to improve the sounds of some parts;
- Section 4.7: lines 353-356: Please provide more citations or explanation for the potential of TGF-CD73-Adenosine axis in relation to CAR-T cells in cancer immunotherapy.
For clarity, in the revised manuscript, this is now Section 4.8.
I have now added additional citations for approaches to target adenosine receptor signalling for CAR-T cell therapies (Refs 97, 98, 99 and 102 in the revised manuscript). I have also added a brief description of a very recent Phase 1 trial investigating the use of bispecific TGFb-CD73 antibodies in solid tumors as follows (Lines 382-385 of revised manuscript).
“Furthermore, a recent Phase 1 study reported that a CD73-TGFb-targeting bifunctional antibody, dalutrafusp alfa, was well-tolerated in patients with advanced solid tumors [103]; further, larger studies will be required to demonstrate clinical efficacy.”
- Section 4.7: line 367: Please explain in more details and in point of mechanistic view the relation between FoxP3 and TGF beta in cancer immunity.
I think the reviewer is referring to FoxP1 rather than FoxP3 in this section - I have added the following sentence to clarify the role of FoxP1 in mediating the inhibitory effects of TGFbeta (lines 389-391 of revised manuscript).
“FoxP1 was shown to be a necessary component of a transcriptionally repressive TGFb-induced SMAD2/3 complex that bound to promoter regions and limited transcription of target genes.”
Reviewer 2 Report
This is an interesting literature on “Regulation of T cell activation and metabolism by transforming 2 growth factor-beta”. Overall review is well written with well focus on T cells associated metabolic pathway regulated by TGFβ, however there are some key consideration that will make this review more informative.
1- Author should also talk about the role of TGFβ signaling and metabolic pathways in some other T cell subtypes such as Th22 and Tfh cells in detail, like he has mentioned about some other T cell phenotype.
2- Author has discussed about the importance of energy metabolic pathways such as glycolytic and oxidative phosphorylation, however it would be more informative if author also discuss recent development about other pathways such as cholesterol and fatty acid metabolism in these setting.
3- In conclusion author should also talk about the recent development in TGFβ and T cell activation as a therapeutic target for human diseases and future potential in clinical settings.
Author Response
I'd like to thank the reviewer for their helpful comments. I have added new text to the manuscript to address the key points as detailed below (reviewers comments are in bold, my replies in italics).
This is an interesting literature on “Regulation of T cell activation and metabolism by transforming 2 growth factor-beta”. Overall review is well written with well focus on T cells associated metabolic pathway regulated by TGFβ, however there are some key consideration that will make this review more informative.
1- Author should also talk about the role of TGFβ signaling and metabolic pathways in some other T cell subtypes such as Th22 and Tfh cells in detail, like he has mentioned about some other T cell phenotype.
The roles of TGFb in the differentiation of Th22 and Tfh cells are described in the following new section, in the revised manuscript Lines 133-144, with additional references 37-42 now cited.
“The role of TGFb in the differentiation of IL-22-secreting T cells has been controversial. Data from human studies suggests that Th22 cells are a stable and distinct helper subset [37] whilst in vitro studies suggested that TGFb limits the polarization of naïve CD4+ T cells to a Th22 phenotype [38]. By contrast, more recently, evidence has shown that TGFb promotes IL-22 production by Th17 cells in vitro and in vivo [39]. Furthermore, a positive role for TGFb signalling in the licensing and differentiation of follicular helper T cells (Tfh), that localize to B cell follicles of secondary lymphoid tissue, has been reported [40, 41]. During CD4+ T cell differentiation, TGFb suppresses expression of the chromatin organizer and transcription factor special AT-rich sequence-binding protein-1 (SATB1) via SMAD3-dependent signals, that in turn enables enhanced expression of inducible co-stimulator (ICOS) that promote Tfh differentiation [41, 42].”
A new citation (Ref 51 in revised manuscript) has been added and the following sentences describing the metabolic regulation of Tfh polarisation added (Lines 178-182):
“Furthermore, recent studies using an in vivo CRISPR screen approach indicate a key role for the regulation of metabolic pathways in the Tfh versus Th1 fate decision. In this study, HIF-1a was shown to suppress Myc and mTORC1-dependent pathways whilst degradation of HIF1a was selectively required for Tfh differentiation [51].”
2- Author has discussed about the importance of energy metabolic pathways such as glycolytic and oxidative phosphorylation, however it would be more informative if author also discuss recent development about other pathways such as cholesterol and fatty acid metabolism in these setting.
Fatty acid/lipid metabolism have been discussed throughout the manuscript (e.g. Line 66, 86. 173-175, 197-200, 222-224, 314-315, 344-349 of the revised manuscript). Some additional text and references have been added to clarify this:
Lines 74-76 (new refs 8,9) – “In both effector and memory T cells glucose metabolism, via glycolysis and the TCA cycle, is also linked to de novo fatty acid synthesis [8, 9].”
Lines 176-178 – “Th17 differentiation conditions induce the upregulation of acetyl-CoA carboxylase 1 (ACC1), that mediates glucose-dependent de novo fatty acid synthesis, whilst deletion of ACC1 diverts Th17 cells to a Treg phenotype.”
3- In conclusion author should also talk about the recent development in TGFβ and T cell activation as a therapeutic target for human diseases and future potential in clinical settings.
As requested, in the conclusion section, I have added an additional paragraph describing some of the most recent preclinical and clinical studies that have assessed targeting TGFb in cancer therapies (Lines 418-439) and cited additional references 111-115.
“In recent years, the efficacy of targeting TGFb in combination with other cancer immunotherapy modalities has been assessed. A number of clinical trials have been established to test the impact of combined TGFb inhibition and immune checkpoint inhibitors in a range of treatment-refractory cancer types (reviewed in [111]). Importantly, a recent phase 1 trial demonstrated that dual PD-L1 and TGFb blockade safely enhanced anti-tumor immunity in a cohort of patients with newly diagnosed human papilloma virus (HPV)-unrelated head and neck squamous cell carcinoma [112]. Furthermore, a recent preclinical study demonstrated that agonists of the TGFb family member bone morphogenetic protein 4 could oppose TGFb-induced effector dysfunction in human T cells and improve response to PD-1 immune checkpoint blockade in mice, suggesting alternative approaches to blocking the inhibitory effects of TGFb in anti-cancer immunity [76]. Recent evidence suggests that combining TGFb blockade with adoptive T cell therapies may also improve cancer outcomes. Thus, studies have reported that combining receptor tyrosine kinase like orphan receptor 1 (ROR1)-specific CAR T cells with the TGFBR kinase inhibitor SD-208 enhanced clearance of tumors in a mouse model of triple-negative breast cancer [113] whilst CRISPR-mediated deletion of TGFBR2 improved the efficacy of anti-mesothelin CAR-T cells in patient-derived pancreatic cancer xenograft models [114]. Furthermore, in-human phase I trials determined the feasibility of targeting TGFBR in prostate cancer-directed CAR T cells, using a dominant-negative receptor approach [115]. Alongside these established approaches to block TGFb signaling, it is hoped that an improved understanding of TGFb-dependent targets, including metabolic pathways, may lead to improved or more selective targets for T cell-based immunotherapies.”